# *Phellinus linteus* Mycelia Extracts Show Potent Antiviral and Immunomodulatory Effects in H1N1 Influenza Virus-Infected Mice

**DOI:** 10.3390/foods14234047

**Published:** 2025-11-26

**Authors:** I-Chen Li, Yi-Lin Chan, Wen-Ting Lu, Lynn-Huey Chiang, Tsung-Ju Li, Tsung-Lin Li, Chin-Chu Chen, Chang-Jer Wu

**Affiliations:** 1Biotech Research Institute, Grape King Bio Ltd., Long Tan District, Taoyuan City 325, Taiwan; ichen.li@grapeking.com.tw (I.-C.L.); lynnhuey.chiang@grapeking.com.tw (L.-H.C.); tsungju.li@grapeking.com.tw (T.-J.L.); 2Department of Chemistry, Chinese Culture University, Taipei 111, Taiwan; phd.elainechan@gmail.com; 3Department of Food Science and Center of Excellence for the Oceans, National Taiwan Ocean University, Keelung City 202, Taiwan; a29405090@gmail.com; 4Genomics Research Center, Academia Sinica, Taipei 115, Taiwan; tlli@as.edu.tw; 5Institute of Food Science and Technology, National Taiwan University, Taipei City 106, Taiwan; 6Institute of BioPharmaceutical Science, National Sun Yat-sen University, Kaohsiung 804, Taiwan; 7Department of Bioscience Technology, Chung Yuan Christian University, Zhong-Li District, Taoyuan City 320, Taiwan; 8Graduate Institute of Medicine, Kaohsiung Medical University, Kaohsiung 807, Taiwan

**Keywords:** *Phellinus linteus* mycelia GKPL, extracts, active compounds, H1N1, immunomodulatory effects

## Abstract

*Phellinus linteus*, a medicinal mushroom with an ancient history in traditional medicine, serves as a valuable raw material for functional foods. While previous studies have shown promising antiviral effects, the present work is the most comprehensive investigation of *P. linteus* mycelial extracts and active compounds thereof against the H1N1 influenza virus infection in in vitro and animal models. Antiviral activity was evaluated using three treatment approaches (preventive, co-treatment, and therapeutic) in MDCK cells infected with H1N1 virus. In vivo studies employed male BALB/c mice exposed to aerosolized H1N1 virus (1.6 × 10^6^ PFU/mL). In the animal experiment, mice received either *P. linteus* mycelia powder (PL) at 1000 mg/kg/day or PLw/PLe at 350 mg/kg/day for 7 days before and after infection. Ethanol extracts (PLe) demonstrated superior antiviral properties compared to water extracts (PLw) in cell survival and viral inhibition tests. Animal studies revealed that both PL and PLe significantly improved recovery after H1N1 infection, with survival rates of 60% and 40%, respectively, versus 25% in controls. The treatments effectively restored depleted immune cell populations, indicating broad immunomodulatory effects. These findings highlight the potential of *P. linteus* as a functional food ingredient with promising therapeutic applications against viral infections through its action on both innate and adaptive immune responses.

## 1. Introduction

Influenza virus, particularly type A, is a serious global concern and a health hazard in its own right due to its ability to cause severe respiratory illness and acute lung injury [1]. The virus’s high mutation rate, efficient transmission, and rapid development of drug resistance pose significant challenges to the current therapeutic modalities [2]. While neuraminidase inhibitors like Tamiflu and Relenza are widely used, their efficacy is limited by various facets of facts, such as adverse effects, emerging resistance, and high costs [3]. These limitations underscore an urgent need for novel anti-influenza strategies and treatments to mitigate influenza A virus-induced lung damage.

*Phellinus linteus*, a fungus with a millennia-long history as a traditional medicine, has recently regained scientific attention for its possession of diverse bioactive compounds [4]. These compounds include phenylpropanoids, terpenoids, and furans carrying phenomenal therapeutic activities against diseases from cancer to viral infections [4]. Extensive in vitro studies have demonstrated that extracts from this fungus possess significant antiviral properties. For instance, Seo et al. showed that polyphenols (hispidin, hypholomine B, inoscavin A, davallialactone, and phelligridin D), sesquiterpenoid, inotilone, and 4-(3,4-dihydroxyphenyl)-3-buten-2-one isolated from *Phellinus* spp. targeted viral neuraminidase [5]. Furthermore, Hwang et al. demonstrated that hispidin and hispidin polyphenols, major bioactive compounds found in *P. linteus* mycelia, non-competitively inhibited the proliferation of influenza virus H1N1, H5N1, and H3N2 [6]. Similarly, Lee et al. reported that Phellinus water extract was effective against influenza A and B viruses, including 2009 pandemic H1N1, human H3N2, avian H9N2, and oseltamivir-resistant H1N1 viruses [7].

Despite these valuable findings, most existing studies have focused on isolated compounds or fruiting-body extracts, leaving a gap in understanding the antiviral potential of mycelium-derived extracts as complete, standardized mixtures. Moreover, previous work rarely compares different extraction fractions or examines whether water and ethanol extracts exert distinct biological activities during influenza infection. Importantly, no prior study has evaluated the protective effects of *P. linteus* mycelial extracts in both in vitro and in vivo influenza A infection models or investigated their ability to modulate host immune signaling and lung injury–related pathways. These gaps highlight the need for a more comprehensive investigation that links extract composition, antiviral activity, and immunomodulatory effects in the context of influenza A infection.

Our previous research has shown that *P. linteus* mycelia (GKPL) and its active compounds hispidin and hypholomin B exhibited promising antiviral effects against COVID-19 [8]. Both COVID-19 and influenza are respiratory viral infections that share common pathogenic mechanisms. For instance, influenza A virus has been reported to modulate ACE2 (angiotensin-converting enzyme 2, a receptor involved in viral entry and regulation of lung injury) expression and contribute to acute lung injury, while downregulation of ACE2 has been closely linked to the severity of COVID-19-associated lung damage [9].

Building from these foundations, our earlier findings demonstrated that hispidin inhibits SARS-CoV-2 pseudovirus infection by downregulating ACE2 gene expression in HepG2 cells, thereby reducing viral entry. In this study, we aimed to bridge the knowledge gap by investigating the protective mechanisms of *P. linteus* mycelium extracts against influenza A virus infection. By focusing on the mycelium rather than the fruiting body, we sought to identify novel bioactive compounds and exploit potential synergistic effects unique to this part of the fungus. Ultimately, our goal was to develop an innovative, nature-derived strategy for combating influenza and its associated lung injury. Our results not only broaden the understanding of *P. linteus*’s medicinal properties but also highlight new opportunities for antiviral drug discovery and development.

## 2. Materials and Methods

### 2.1. Sample Preparation

*P. linteus* specimens used in this study were obtained from wild fruit bodies harvested in the mountainous region of Guanxi Township (Hsinchu, Taiwan). The specimens were authenticated by Genomics Company in Taiwan and subsequently deposited at the Bioresource Collection and Research Center (BCRC) with an accession number 930210. The mycelia, initially grown on potato dextrose agar (PDA), were transferred to a 1 L optimized broth (1% fructose, 0.3% yeast extract, and 0.05% MgSO_4_, pH 4.5) in a 2 L flask. They were then cultivated on a rotary shaker at 120 rpm and 25 °C for one week. The fermentation process was subsequently scaled up from a shake flask to a 500 L fermenter, and finally to a 40-ton fermenter for 10 days. After cultivation, the mycelia were harvested, lyophilized, ground into powder, and stored in a desiccator at room temperature for further extraction.

*P. linteus* mycelial freeze-dried powder (PL) was further extracted into water and ethanol extracts by mixing with pure water and ethanol at a 1:20 weight ratio. The water mixture was heated to 121 °C for 30 min, while the ethanol mixture was sonicated in an ultrasonic bath for 1 h. Both mixtures were then filtered through Whatman filter paper No. 4 and concentrated using a rotary evaporator (R-220; Büchi Labortechnik AG, Flawil, Switzerland). This process yielded two products: a hot water extract (*P. linteus* mycelia water extract, PLw) and an ethanol extract (*P. linteus* mycelia ethanol extract, PLe). The dried extracts were then redissolved in water and DMSO, respectively, to a final concentration of 100 mg/mL. Lastly, they were filtered through a sterilized 0.22 μm syringe in preparation for subsequent testing.

### 2.2. Cell Culture and Viral Preparation

For cell culture, Madin–Darby canine kidney (MDCK) cells (ATCC CCL) were cultured in Dulbecco’s modified Eagle’s medium (DMEM; Gibco, Grand Island, NY, USA) supplemented with 10% fetal bovine serum (FBS; Gibco) at 37 °C in a humidified 5% CO_2_ incubator. The influenza A virus A/WSN/33 (H1N1) strain was propagated in specific pathogen-free (SPF) chicken embryos (JD-SPF Biotech Co. Ltd., Miaoli County, Taiwan). Briefly, 200 μL of influenza virus solution (10^5^ PFU/mL) was inoculated into the allantoic cavity through the chorioallantoic membrane. After sealing with paraffin, eggs were incubated at 37 °C for 48 h, followed by chilling at 4 °C for 4 h. The allantoic fluid was then harvested and stored at −20 °C [10]. Virus titers were determined by plaque assay, and virus stocks were stored at −80 °C until use.

### 2.3. Virus Infection In Vitro

The antiviral activity of PLw and PLe was evaluated using MDCK cells. Cell viability was first assessed using the MTS assay, which measures metabolic activity as an indicator of viable cells [11] (Appendix A), and non-cytotoxic concentrations were selected for subsequent experiments. For this assay, MDCK cells (1.5 × 10^4^ cells/well) were seeded in 96-well plates and incubated for 24 h to allow attachment, then treated with various concentrations of PLw (Appendix A) and PLe (Appendix A) for 48 h. After treatment, MTS reagent was added at a 5:1 ratio of culture medium to reagent and incubated for 1 h in the dark, and absorbance was measured at 490 nm using an Agilent BioTek Epoch 2 Microplate Spectrophotometer (Santa Clara, CA, USA) to determine cell viability.

For antiviral testing, three treatment strategies were:Pre-treatment: MDCK cells were treated with PLw and PLe at various concentrations for 1 h prior to H1N1 infection (MOI = 0.1) for an additional hour.Simultaneous treatment: H1N1 virus was pre-incubated with different concentrations of PLw and PLe for 1 h before cell infection (1 h duration).Post-infection treatment: Following H1N1 infection (MOI = 0.1, 1 h), cells were exposed to varying concentrations of PLw and PLe for 1 h.

After the respective treatments, the medium was replaced with DMEM supplemented with 2% FBS, and plates were incubated at 37 °C in 5% CO_2_ for 48 h. Cell viability was subsequently determined using the MTS assay.

### 2.4. Plaque Reduction Assay

The plaque reduction assay was performed to evaluate the antiviral activity. MDCK cells were seeded at 2.5 × 10^5^ cells/well in 6-well plates and categorized into three treatment groups: (a) Preventive, (b) Co-treatment, and (c) Therapeutic. The cells were infected with 50 PFU/well of H1N1 virus and incubated at 37 °C with 5% CO_2_ for 24 h. Following treatment, the medium was replaced with DMEM containing 2% FBS and 2% SeaPlaque agarose. After solidification, plates were incubated at 37 °C with 5% CO_2_ for 48 h. The cells were then fixed with 1 mL/well of 10% formalin for 1 h at room temperature. After gel removal, cells were stained with 1% crystal violet for 1 h. Plaques were counted following washing to determine the virus titer (PFU/mL). The viral plaque formation inhibition rate was calculated as: (%) = [1 − (plaques in test group/plaques in vehicle control group)] × 100%. All experiments were conducted in triplicate.

### 2.5. Virus Infection In Vivo

All animal experimental procedures and animal welfare protocols were approved by the Laboratory Animal Center ethics committee (authorization number: 109073), National Taiwan Ocean University, Taiwan. Male BALB/c mice (5–8 weeks old, 20–26 g) were obtained from the National Laboratory Animal Center (Taipei, Taiwan) and maintained in a BSL-2 facility at the Animal Experimental Center, College of Life Sciences, National Taiwan Ocean University. Animals were housed in cages under standard laboratory conditions (12-h light/dark cycle, temperature 24 °C ± 2 °C, 50–70% humidity) with unrestricted access to LabDiet 5001 rodent feed and purified water.

Mice were randomly divided into the following groups: control group (*n* = 8, non-infected), H1N1-infected group (*n* = 8), Tamiflu treatment group (10 mg/kg/day post-infection, *n* = 8), PL treatment group (1000 mg/kg/day, *n* = 10), and PLe and PLw treatment groups (350 mg/kg/day each, *n* = 10). Randomization was performed using Random Number Generator (https://www.calculatorsoup.com/calculators/statistics/random-number-generator.php, accessed on 16 May 2022) to ensure unbiased group allocation. The sample size was informed by previous H1N1 infection studies in mice [12], in which changes in body weight and survival rate served as primary outcomes and demonstrated clear treatment effects with 8–10 animals per group. Accordingly, eight mice were assigned to the control and infection groups, and ten mice to each treatment group to ensure adequate statistical reliability and account for biological variability.

Treatment administration began 7 days pre-infection and continued for 7 days post-infection via daily oral gavage, except for the Tamiflu group, which received treatment only post-infection. Virus infection was administered via aerosol exposure to 1.6 × 10^6^ PFU/mL (LD_50_) for 30 min, while controls received phosphate-buffered saline (PBS). Half of each group was sacrificed on day 7 post-infection for collection of blood, lung tissue, and bronchoalveolar lavage fluid (BALF). The remaining animals were monitored until day 14. Body weight changes and survival rates were recorded daily to assess treatment efficacy. Animals showing severe clinical signs were humanely euthanized according to IACUC guidelines.

### 2.6. Hematology Analysis

Animals were euthanized using a gradual-fill CO_2_ method following AVMA guidelines. The gas was introduced at a 30–70% chamber volume displacement rate per minute without prefilling the chamber. Animals were monitored until respiratory arrest occurred, with CO_2_ flow maintained for at least one minute after breathing ceased to confirm death. Following euthanasia, cardiac blood was collected from the mice into heparin-containing tubes for subsequent analysis. The blood was then analyzed using an Exigo hematology analyzer (Medonic, Sweden) to determine white blood cell (WBC) counts. Blood samples were further centrifuged at 3000× *g* for 30 min. The resulting supernatant serum was used to measure IL-6 (Affymetrix, Santa Clara, CA, USA) and IFN-γ (Invitrogen, Carlsbad, CA, USA) levels using ELISA kits, following the manufacturer’s instructions.

### 2.7. BALF Analysis

After euthanasia, 800 μL of PBS was injected into the lungs through the trachea using a 25G needle and washed back and forth twice. The fluid was then collected in a 1.5 mL microcentrifuge tube and centrifuged at 3000× *g* for 30 min. The resulting supernatant was used to measure IL-6 (#BMS603-2; Thermo Fisher Scientific, Chicago, IL, USA) and IFN-γ (MIF00; R&D Systems, Minneapolis, MN, USA) levels using ELISA kits, according to the manufacturer’s instructions. Meanwhile, the cell pellet was analyzed using flow cytometry.

### 2.8. Flow Cytometry

To analyze BALF samples, the cell pellet was first treated with 300 μL RBC lysis buffer to eliminate red blood cells. After centrifugation, the cells were stained with BD Biosciences fluorescent dyes (FITC, PE, and PE-Cy5) under controlled conditions. Flow cytometry was then employed to determine surface marker ratios. Lymphocyte subsets in BALF were examined using dual flow cytometry (FACSscan, Becton-Dickinson Inc., San Jose, CA, USA), focusing on CD3+, CD4+, CD8+, and NK1.1 subsets. Various antibody combinations were used to identify specific cell types, including T cells, helper T-cells, suppressor/cytotoxic T-cells, and NK-cells. The comprehensive analysis included over 5000 cells from the lymphocyte region. Results were expressed both as a percentage of lymphocytes and as cell counts per 10^3^/mL, providing a thorough characterization of the BALF lymphocyte population.

### 2.9. Real-Time PCR

RT-qPCR was performed to analyze gene expression in mouse lung tissue. Total RNA was extracted from 25–30 mg of right lung tissue using RNeasy Mini Kit (Qiagen, Hilden, North Rhine-Westphalia, Germany) according to the manufacturer’s protocol. RNA was reverse transcribed to cDNA using Oligo(dT) primer, M-MLV 5X Reaction Buffer, dNTP mix (10 mM), RNaseOUT, and M-MLV Reverse Transcriptase (Thermo Fisher Scientific, USA). The reaction conditions were 42 °C for 60 min, followed by 95 °C for 5 min. Real-time PCR was performed using qPCRBIO SyGreen Mix (PCR Biosystems, Boston, MA, USA) with gene-specific primers (Appendix A). GAPDH was used as the internal reference gene. The relative gene expression was calculated using the 2^−ΔΔCt^ method. All experiments were performed in triplicate.

### 2.10. Pathological Analysis

Following euthanasia by CO_2_ asphyxiation, mouse lungs were extracted, weighted and immersed in 10% formalin solution at 4 °C for 24 h. The tissues were then dehydrated using a series of ethanol solutions (70%, 90%, 100%) and xylene, before being embedded in paraffin. Tissue sections were cut and placed on slides, which were then heated in a 50 °C oven and stored at 4 °C or prepared for staining. For staining, paraffin was removed from the slides using xylene, and the tissues were rehydrated. The slides were then stained with hematoxylin for 10 s, washed with running water for 15 min, and counterstained with eosin for 1 min. After washing, the slides were dehydrated again using a series of ethanol solutions (70%, 90%, 100%) and xylene for 5 min each. Once dry, the slides were coverslipped for preservation and analysis using a light microscope (Olympus BX-50, Tokyo, Japan) to examine pathological changes.

### 2.11. Statistical Analysis

Results are expressed as mean ± standard deviation (SD). Data were first assessed for normality using the Shapiro–Wilk test and for homogeneity of variance using Levene’s test. Data analysis was conducted using GraphPad Prism version 8.0. For multiple group comparisons, one-way ANOVA was employed, followed by Dunnett’s post hoc test. A *p*-value less than 0.05 was considered statistically significant.

## 3. Results

### 3.1. Effects of PLw and PLe Against H1N1 Infection in MDCK Cells

The cytotoxicity assay was performed in the first place to evaluate the effects of *P. linteus* mycelial extracts on MDCK cells prior to antiviral experiments. As shown in Appendix A, PLw shows no significant cytotoxicity up to 1000 μg/mL, where the cell viability is still above 80% (*p* > 0.05). The estimated CC_50_ value for PLw was therefore greater than 1000 μg/mL, indicating low cytotoxicity. In contrast, PLe exhibited cytotoxicity at higher concentrations of 1000 and 500 μg/mL (*p* < 0.05; Appendix A), with an estimated CC_50_ of 577 μg/mL. As a result, 1000 μg/mL of PLw and 250 μg/mL of PLe were opted for and subjected to subsequent antiviral studies.

Next, the antiviral effects of *P. linteus* mycelial extracts were evaluated against MDCK cells infected with H1N1 influenza virus (MOI = 0.1) (Figure 1). The infected cells were treated with the extracts under three different conditions: preventive (Figure 1A,D), co-treatment (Figure 1B,E), and therapeutic (Figure 1C,F).

As shown in Figure 1D, PLe at 250 μg/mL demonstrates a significant antiviral activity in the preventive condition, where cell viability is increased by 10.58%. In the co-treatment condition, both extracts show protective effects against viral infection. PLw at concentrations of 250–1000 μg/mL significantly reduces viral replication, as evidenced by increased cell viability of 4.72–5.08% (Figure 1B). PLe exhibits a relatively high potent antiviral activity at concentrations of 62.5–250 μg/mL with increased cell viability by 8.23–18.62% (Figure 1E). In the therapeutic condition, PLe at 125 and 250 μg/mL effectively inhibits viral replication as manifested by increased cell viability by 6.71% and 14.64%, respectively (Figure 1F). These findings indicate that PLe possesses broad antiviral activity across all treatment conditions, with brilliant antiviral effects, particularly proven by the co-treatment. PLw demonstrates a moderate antiviral activity primarily shown at the co-treatment.

### 3.2. Effects of PLw and PLe on Plaque Formation in MDCK Cells Infected by H1N1

The plaque reduction assay was conducted to evaluate the antiviral effects of PLw and PLe extracts against the H1N1 virus (Figure 2 and Appendix A). As shown in Figure 2B, PLw that exhibits antiviral activity only occurs at the co-treatment condition, where the plaque reduction rates are about 21.62–45.95% for concentrations between 125–1000 μg/mL. Since 50% inhibition was not achieved even at the highest concentration tested, the estimated EC_50_ for PLw is greater than 1000 μg/mL. In contrast, PLe demonstrates more potent antiviral effects across all treatment conditions. In the preventive treatment, PLe at 250 μg/mL achieves 32.26% of plaque reduction (Figure 2D). The highest antiviral activity is occurred at the co-treatment condition with application of PLe, showing 39.52–69.35% plaque reduction at 31.25–250 μg/mL (Figure 2E). Based on this data, the estimated EC_50_ for PLe is approximately 118 μg/mL. In the therapeutic treatment, PLe reduces plaque formation by 24.20% and 31.21% at 125 and 250 μg/mL, respectively (Figure 2F). Added together, our results indicate that PLe effectively inhibits H1N1 viral replication but with an optimal effect for the co-treatment.

### 3.3. Effects of PL, PLw and PLe on Survival Rate and Body Weight in H1N1-Infected Mice

There is a need to determine the optimal H1N1 infection dose prior to performing animal-model experiments. Mice were exposed to various virus concentrations (1 × 10^4^ to 1 × 10^7^ PFU/mL) via aerosol inhalation for 30 min (Appendix A). The LD_50_ 1.6 × 10^6^ PFU/mL was thereby established using the Spearman–Kärber method [13], which was used in subsequent experiments to ensure a consistent disease model.

Body weight changes and survival rates were monitored over 14 days post-infection. The untreated H1N1 control group showed progressive weight loss starting at day 3 with a decreased survival rate of 75% at day 10 and dropping down to 25% at day 14 (Figure 3A). In contrast, the Tamiflu-treated group (positive control) maintained a 100% survival rate throughout the experiment (Figure 3B). The PL and PLe treatment groups demonstrated moderate protection with mice showing recovery at day 9 (significantly earlier than the H1N1 control, *p* < 0.05; Figure 3A). Both PL and PLe groups maintained a 60% survival rate at day 11, though that of PLe declined to 40% at day 14. The PLw group showed the least efficacy, with a survival rate of 80% at day 10 and of 20% at day 14.

### 3.4. Effects of PL, PLw and PLe on Lung Injury in H1N1-Infected Mice

To better understand the disease progression, lung tissue biopsies were obtained and examined to assess pulmonary inflammation and pathological changes at day 6 post-infection. The control and Tamiflu-treated groups maintained healthy pink lung coloration, while H1N1-infected mice displayed severe congestion and dark red discoloration. By the same token, the PLw-treated lungs exhibited dark red coloration, though those of the PL and PLe treatments demonstrated reduced hemorrhaging in comparison to those of the virus group (Figure 4A).

To quantify lung inflammation, the ratio of lung weight to body weight was calculated, whereby an elevated ratio indicated an increased inflammatory response. The control and Tamiflu groups showed significantly reduced inflammation when compared to the H1N1-infected mice (2.41 and 1.77-fold decrease, respectively; *p* < 0.001). Both the PL and PLe treatments significantly attenuated lung inflammation (*p* < 0.05), while the PLw treatment showed no significant protective effect (*p* > 0.05; Figure 4B).

Moreover, histopathological examination of H&E-stained lung sections for the control group showed a typical alveolar architecture with characteristic single-layer cells. In contrast, the H1N1-infected lungs displayed severe tissue damage as manifested by disrupted alveolar structure, extensive inflammatory cell infiltration, and severe hemorrhage. In general, all treatments can intervene in the viral infection but with varying degrees of protection given the H1N1-induced pulmonary damage (Figure 4C).

### 3.5. Effect of PL, PLw or PLe on Leukocyte Populations in H1N1-Infected Mice

To evaluate immunological responses upon administration of PLs, white blood cell (WBC) counts were analyzed in blood samples collected from mice at day 6 post-infection (Figure 5). The H1N1-infected group showed significantly reduced total WBC counts as opposed to the control mice (Figure 5A). Treatment with *P. linteus* extracts, however, demonstrated varying degrees of WBC reduction, a reflection of the PL extract’s protective effects. Namely, PL extracts showed the strongest immunomodulatory effect, increasing WBC counts by 80.8% compared to the H1N1-infected group (*p* < 0.05; Figure 5A), thereby outperforming Tamiflu (61.3% increase, *p* < 0.05). The PLe treatment resulted in a 49.7% increase in WBC counts (*p* < 0.05), while PLw showed a modest increase, as it was not statistically significant.

Analysis of leukocyte subpopulations revealed that H1N1 infection significantly decreased lymphocyte counts (*p* < 0.05; Figure 5B). Both Tamiflu and PL treatments effectively prevented this decrease (*p* < 0.05; Figure 5B), while both PLw and PLe showed a trend toward lymphocyte recovery in spite of not reaching statistical significance. Among the treatment groups, only the PL extract significantly affected the granulocyte level (*p* < 0.05; Figure 5C), in contrast to other treatments, which showed no significant differences.

### 3.6. Effect of PL, PLw or PLe on Inflammatory Cytokines in H1N1-Infected Mice

To investigate the anti-inflammatory effects of PLs on H1N1-infected mice, inflammatory cytokine levels in blood and bronchoalveolar lavage fluid (BALF) samples were analyzed using ELISA. Blood analysis revealed that H1N1 infection significantly elevated IL-6 and IFN-γ levels to 99.12 pg/mL and 683.52 pg/mL, respectively (*p* < 0.001; Figure 6A). The PL treatment effectively suppressed cytokine responses with reduction in IL-6 to 37.67 pg/mL and that of IFN-γ to 450.42 pg/mL (*p* < 0.05; Figure 6A). Tamiflu demonstrated significant reductions, while the PLw and PLe treatments showed modest but non-significant decreases (*p* > 0.05; Figure 6A). In BALF samples, the control mice showed undetectable levels of both cytokines, whereas H1N1 infection showed a dramatic increase in IL-6 (1146.11 pg/mL) and IFN-γ (698.71 pg/mL) (*p* < 0.001; Figure 6B). As a result, the PL treatment significantly attenuated the cytokine levels to 679.20 and 469 pg/mL, respectively (*p* < 0.05; Figure 6B), while other treatments showed no significant effects.

### 3.7. Effect of PL, PLw or PLe on Gene Expression of the Lung Tissue in H1N1-Infected Mice

To probe the molecular mechanism underlying the anti-inflammatory effects of *P. linteus* extracts, RT-PCR analysis was performed for lung tissue samples. Intriguingly, the analysis showed significant upregulation of both viral and inflammatory markers post-infection of H1N1. Amid the treatments, PL demonstrated the most robust effects, significantly reducing viral HA expression (*p* < 0.05; Figure 7A), while PLe showed modest reductions. Analysis of inflammatory markers revealed that all treatments can effectively suppress the expression of TNF-α (*p* < 0.05; Figure 7B). However, only the PL treatment achieved a significant reduction in IL-6 expression (*p* < 0.05; Figure 7C), while PLw and PLe showed a moderate but non-significant effect. Similarly, all treatments were shown in a position to reduce IFN-γ gene expression to some degree, with only the PL and PLe treatments exhibiting significant suppression when compared to the virus-only group (*p* < 0.05; Figure 7D).

## 4. Discussion

Our investigation into the antiviral activity of *P. linteus* mycelial extracts revealed multiple mechanisms of action that execute excellently against the H1N1 influenza virus infection. We opted for MDCK cells as our primary experimental model because of their well-documented sensitivity to influenza viruses and capability for supporting viral replication [14]. The ethanol extract (PLe) demonstrated consistently superior performance to the water extract (PLw) across multiple indicative parameters, namely, the cell survival and viral inhibition.

For preventive applications, PLe significantly enhanced the cell survival rate by 10.58% (*p* < 0.01; Figure 1D) as well as achieved a notable plaque inhibition rate of 32.26% (*p* < 0.01; Figure 2D). These results highlight PLe’s capabilities of both strengthening cellular defense and preventing initial viral attachment to host cells [15]. The most compelling results observed during co-treatment experiments were that PLe exhibited remarkable efficacy with an 18.62% improvement in cell survival (*p* < 0.001; Figure 1E) and an impressive 69.35% plaque inhibition rate (*p* < 0.001; Figure 2E). These phenomenal effects can be attributed to PLe’s capability to compete with viral hemagglutinin for host cell binding sites or its direct viral inactivation properties [16]. Previous studies have reported that the *P. linteus* methanolic extract specifically inhibits glycoprotein trafficking rather than affecting glycoprotein synthesis [17], although this specified mechanism requires additional studies to validate.

For therapeutic applications, PLe exhibited significant protective effects as manifested by a 14.64% increase in cell survival (*p* < 0.001; Figure 1F) and a 31.21% reduction in plaque formation (*p* < 0.01; Figure 2F). These results strongly suggest that Ple carries a potent neuraminidase inhibition property [18], a crucial mechanism given the neuraminidase’s role in facilitating viral release and spread through the cleavage of terminal sialic acid residues on host cell surfaces. A groundbreaking study by Hwang et al. [6] who identified five key phenolic compounds from the *P. baumii* fruiting body ethanol extract: hispidin, hypholomine B, inoscavin A, davallianlactone, and phelligridin D. All these compounds demonstrated significant neuraminidase inhibitory activities and were capable of reducing virally induced cytopathic effects, with phelligridin D outweighing others. Furthermore, additional compounds isolated from the *P. linteus* fermentation broth, namely inotilone and 4-(3,4-dihydroxyphenyl)-3-buten-2-one, demonstrated dual functionalities in both H1N1 neuraminidase inhibition and antiviral effects in cytopathic reduction assays [19]. The presence of these bioactive compounds in PLe was confirmed through detailed QTof analysis (Appendix A), providing a clear mechanistic explanation for its superior performance to the water extract.

Following these promising in vitro results, we expanded our investigation into animal studies in the hope of substantiating the therapeutic applications of these extracts. Although PLw showed negligible antiviral activity in vitro, we conducted animal studies because there are documented cases where aqueous plant extracts demonstrated protective effects in animal models via immune system modulation or metabolite production, despite showing little or no direct antiviral activity in laboratory settings [20]. Based on safety studies that *P. linteus* mycelium was tolerated as high as a dose of 5 g/kg in ICR mice [21] and that the extraction yield was about 35%, the doses of 1000 mg/kg for *P. linteus* mycelia powder and 350 mg/kg for the extracts were opted for. These doses were further supported by preliminary in vivo experiments conducted in our laboratory, in which lower doses produced limited or inconsistent antiviral effects, whereas the chosen doses demonstrated measurable improvements in clinical symptoms and survival outcomes. Our experimental results were rather encouraging: the mice treated with PL or PLe showed significant recovery from the H1N1 infection at day 9 when compared to both the untreated H1N1 group and the PLw treatment group. This enhanced recovery was corroborated by higher survival rates, where the PL and PLe groups achieved, respectively, 60% and 40% survival rates when compared to 25% of the H1N1 control group. A limitation of this study is that only male BALB/c mice were used; although our past experiments indicated similar responses to H1N1 infection in both sexes, future studies including female mice are warranted to confirm the generalizability of these findings.

Our analysis on immune cell populations revealed significant impacts of H1N1 infection on white blood cell populations in line with clinical observations, where 64.2% of patients developed leukopenia and 35.7% experienced lymphocytopenia [22]. In our mouse model experiments, both the PL and PLe treatments demonstrated remarkable efficacy in restoring these depleted blood cell counts back to their normal levels (*p* < 0.05; Figure 5). Of particular significance was PL’s unique ability to enhance granulocyte numbers, which are essential immune cells equipped with specialized enzyme-filled granules crucial for mounting effective inflammatory responses and combating viral infections [23].

Further investigation revealed that H1N1’s impact on lymphocytes involves multiple complex mechanisms, including direct cellular death, reduced lymphocyte production, disrupted cellular trafficking patterns, and enhanced inhibitory signaling pathways [24]. Our detailed flow cytometry analysis for bronchoalveolar lavage fluids (BALF) revealed significant reductions in both helper and cytotoxic T cell populations (Appendix A), in agreement with previous findings [25]. Notably, treatment with the PL extracts displayed broad immunomodulatory effects, not only increasing NK cell populations but also restoring T cell populations to baseline levels (Appendix A).

The therapeutic efficacy of PL against H1N1 infection appears to be mediated through a sophisticated immunological cascade. We observed that increased lymphocyte populations strongly correlate with enhanced immune system activation in light of effective viral clearance. The process commences when H1N1 infects infected cells, triggering an innate immune response through the secretion of type 1 interferons (IFN-α and IFN-β), which subsequently activate natural killer (NK) cells [26]. These NK cells function as a critical first line of defense, rapidly identifying and eliminating virus-infected cells.

The adaptive immune response then proceeds through two distinct but complementary pathways. In the cell-mediated response, viral antigens presented via MHC class I molecules lead to the activation of CD8+ cytotoxic T lymphocytes, which employ sophisticated perforin-mediated cytotoxicity mechanisms to eliminate infected cells [26]. Simultaneously, antigen presentation through MHC class II molecules triggers the differentiation of CD4+ helper T cells into T helper 1 cells, which orchestrate the immune response by producing key cytokines, including IL-2 and IFN-γ [27]. IL-2 enhances cytotoxic T lymphocyte activity, while IFN-γ promotes B cell differentiation into antibody-producing plasma cells [28].

The effectiveness of this coordinated immune response has been clearly demonstrated by the significant reduction in hemagglutinin (HA) expression, particularly notable in the PL treatment group. We propose that PL’s superior efficacy compared to either the PLe or PLw extracts is attributable to its comprehensive profile of bioactive compounds, which may act synergistically to modulate multiple immune pathways simultaneously. However, this hypothesis requires further validation through detailed cytokine profiling studies to fully elucidate the precise immunological mechanisms involved.

The role of inflammatory cytokines in viral lung infections, particularly TNF-α, IL-1β, IL-6, and IL-10, has been well-documented as a key factor in determining disease severity, especially in pediatric cases [29]. Clinical studies involving fifty-seven adult patients with confirmed H1N1 infection revealed significantly elevated levels of TNF-α and IL-6 in fatal cases [30]. Additionally, H1N1 virus infection typically induces increased levels of IFN-γ [31], which plays a crucial dual role in both preventing viral replication and promoting both innate and adaptive immune responses [32]. Consistent with previous studies, our H1N1-infected group showed elevated levels of TNF-α, IL-6, and IFN-γ in serum, BALF, and lung tissues. However, groups treated with PL showed significant decreases in all these inflammatory markers, with PL demonstrating greater efficacy than PLe, likely due to its more comprehensive profile of bioactive compounds. These results strongly indicate PL’s superior ability to alleviate inflammation.

Despite the promising results, several limitations should be acknowledged. First, potential sources of bias include the lack of blinding during treatment administration and outcome assessment, which may have influenced subjective measurements. Second, the H1N1 mouse model, while widely used, does not fully recapitulate the complexity of human influenza infection, and extrapolation of efficacy to humans should be approached cautiously. Third, sample sizes, although consistent with prior studies, were relatively small, which may limit statistical power and contribute to variability in observed effects. Future studies incorporating larger cohorts, multiple animal models, and more detailed mechanistic analyses will be essential to validate these findings and clarify the translational potential of *P. linteus* extracts.

## 5. Conclusions

The present work demonstrates that *P. linteus* mycelia effectively act on H1N1 influenza virus, of which ethanol extracts (PLe) showed a better antiviral activity than water extracts (PLw) at both cellular and animal levels. Notably, in the H1N1-infected mouse model, PL and PLe treated groups improved their survival rates, respectively, to 60% and 40% when compared to the control group’s 25%. The extract acts through multiple immune pathways, including blood cell count restoration, NK cell enhancement, and T cell regulation. Importantly, the PL administration reduced key inflammatory markers (TNF-α, IL-6, and IFN-γ) across sera, bronchoalveolar lavage fluids, and lung tissues. In brief, we conclude that the multi-compounds of *P. linteus* exert a synergistic mode of action, substantiating itself as a valuable candidate for future influenza treatments.

## Figures and Tables

**Figure 1 foods-14-04047-f001:**
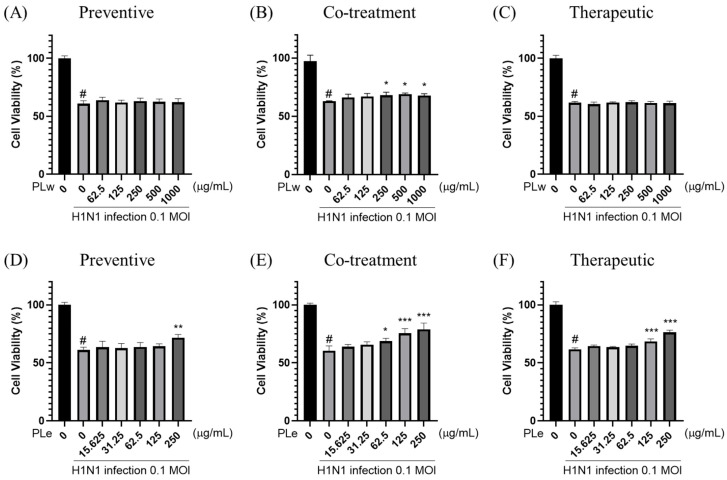
Evaluation of antiviral activity of *P. linteus* mycelial extracts against influenza A virus H1N1. MDCK cells were infected with H1N1 virus (MOI 0.1) and treated with either water extract (PLw; **A**–**C**) or ethanol extract (PLe; **D**–**F**) under three experimental conditions: (1) Pre-treatment: MDCK cells were treated with PLw and PLe at various concentrations for 1 h prior to H1N1 infection (MOI = 0.1) for an additional hour (**A**,**D**); (2) Simultaneous treatment: H1N1 virus was pre-incubated with different concentrations of PLw and PLe for 1 h before cell infection for 1 h (**B**,**E**); (3) Post-infection treatment: Following H1N1 infection (MOI = 0.1, 1 h), cells were exposed to varying concentrations of PLw and PLe for 1 h (**C**,**F**). After the respective treatments, the medium was replaced with DMEM supplemented with 2% FBS, and plates were incubated at 37 °C in 5% CO_2_ for 48 h. Cell viability was subsequently determined using the MTS assay. Values represent mean ± SD from three independent experiments. # *p* < 0.05 vs. untreated control; * *p* < 0.05, ** *p* < 0.01, *** *p* < 0.001 vs. virus control.

**Figure 2 foods-14-04047-f002:**
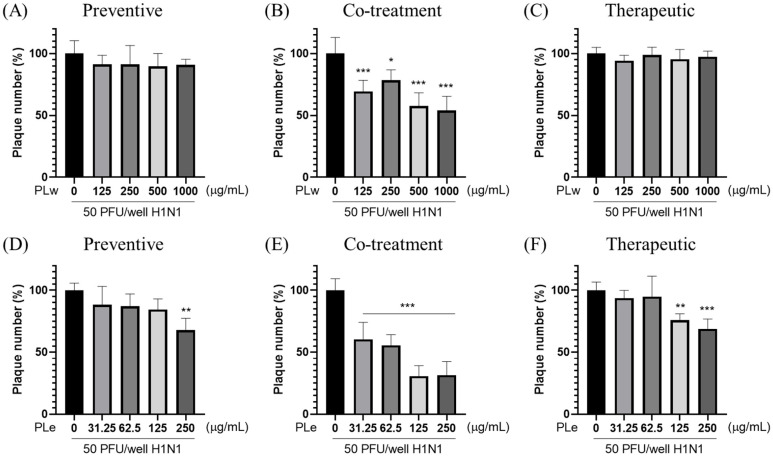
*P. linteus* extracts inhibit influenza A virus H1N1 plaque formation in MDCK cells. The inhibitory effects of water extract (PLw; **A**–**C**) and ethanol extract (PLe; **D**–**F**) were evaluated under different treatment conditions: preventive (**A**,**D**), co-treatment (**B**,**E**), and therapeutic approaches (**C**,**F**). MDCK cells were infected with H1N1 virus and treated with varying concentrations of extracts. Plaque formation was assessed after 48h of incubation. Data represent mean ± SD from three independent experiments. * *p* < 0.05, ** *p* < 0.01, *** *p* < 0.001 vs. virus control.

**Figure 3 foods-14-04047-f003:**
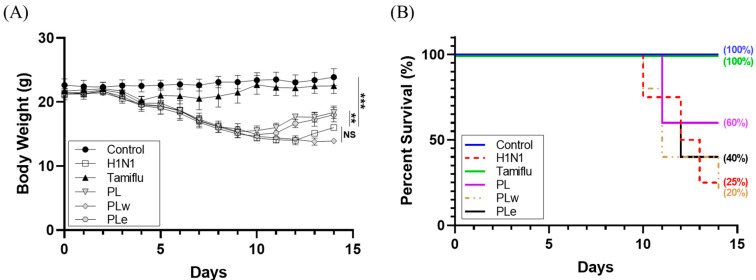
BALB/c mice were infected with A/WSN/33 (H1N1) virus (1.6 × 10^6^ PFU/mL) via aerosol exposure. Treatment groups received daily doses of PL (1000 mg/kg), PLw (350 mg/kg), or PLe (350 mg/kg) for 7 days before and after viral challenge. The positive control group received Tamiflu (10 mg/kg/day) for 7 days post-infection. Mice were monitored for (**A**) body weight changes and (**B**) survival over 14 days. Values are expressed as mean ± SD (n = 8 for control, H1N1, and Tamiflu groups; n = 10 for PL, PLw, and PLe groups). Statistical significance: ** *p* < 0.01, *** *p* < 0.001 versus virus-only group.

**Figure 4 foods-14-04047-f004:**
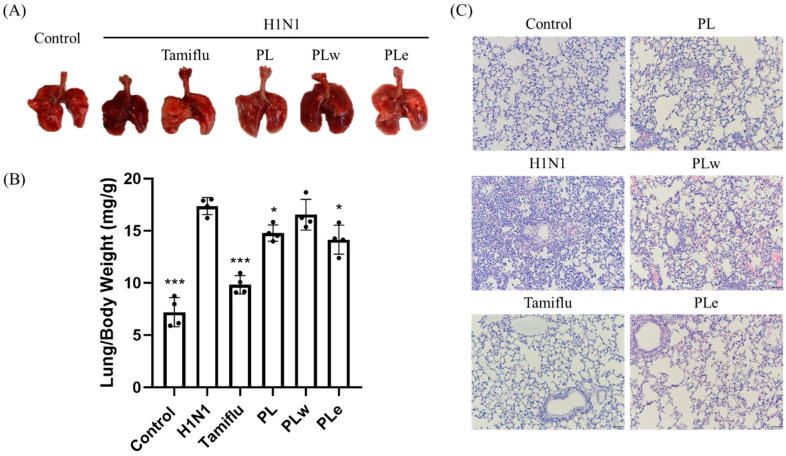
*P. linteus* treatment ameliorates H1N1-induced lung pathology in mice. Mice were administered PLw, PLe, or control treatments followed by H1N1 infection. Lung tissues were collected at day 6 post-infection for analysis. (**A**) Representative gross morphological changes in lung tissues. (**B**) Quantitative analysis of lung inflammation via lung/body weight ratio. (**C**) Histopathological examination of lung sections by H&E staining. Data are expressed as mean ± SD (n = 4). Statistical significance: * *p* < 0.05, *** *p* < 0.001 versus virus-infected group. Magnification: 200x.

**Figure 5 foods-14-04047-f005:**
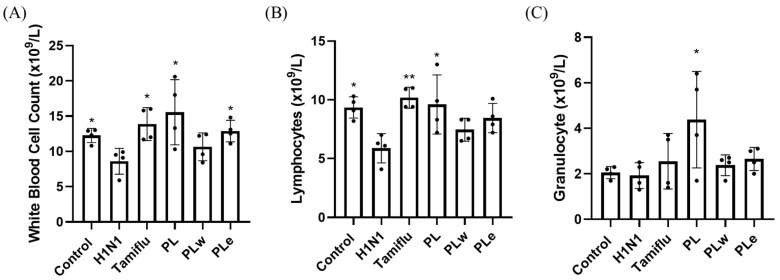
*P. linteus* ameliorated H1N1-induced changes in peripheral blood leukocytes. Blood samples were collected from mice (n = 4) and analyzed for (**A**) total white blood cell (WBC), (**B**) lymphocyte, and (**C**) granulocyte counts using an automated blood cell analyzer. Data are presented as mean ± SD. * *p* < 0.05, ** *p* < 0.01 compared to the H1N1-infected group.

**Figure 6 foods-14-04047-f006:**
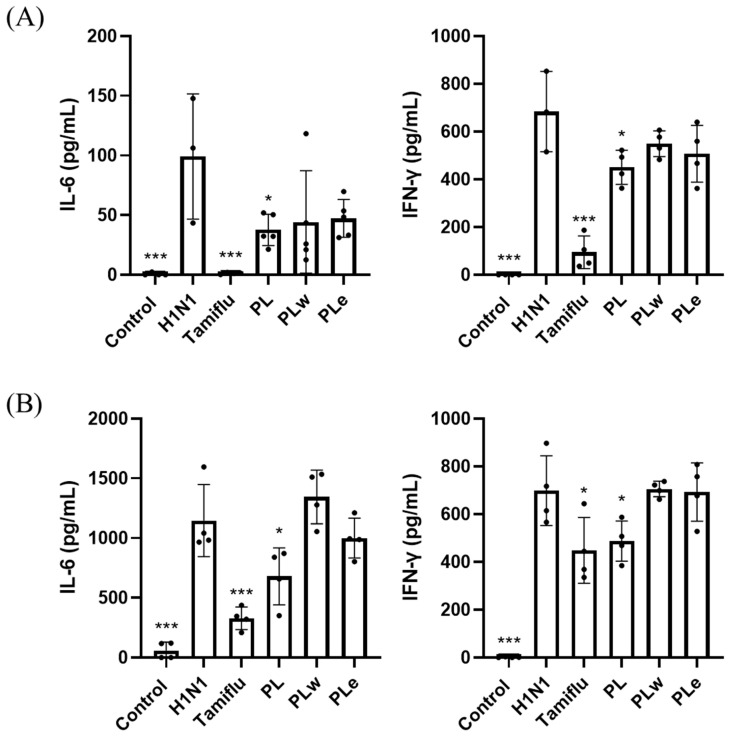
Effects of *P. linteus* extracts on inflammatory cytokine levels in H1N1-infected mice. IL-6 and IFN-γ levels were quantified in (**A**) serum and (**B**) bronchoalveolar lavage fluid by ELISA. Data shown as mean ± SD (n = 4). * *p* < 0.05, *** *p* < 0.001 vs. H1N1-infected group.

**Figure 7 foods-14-04047-f007:**
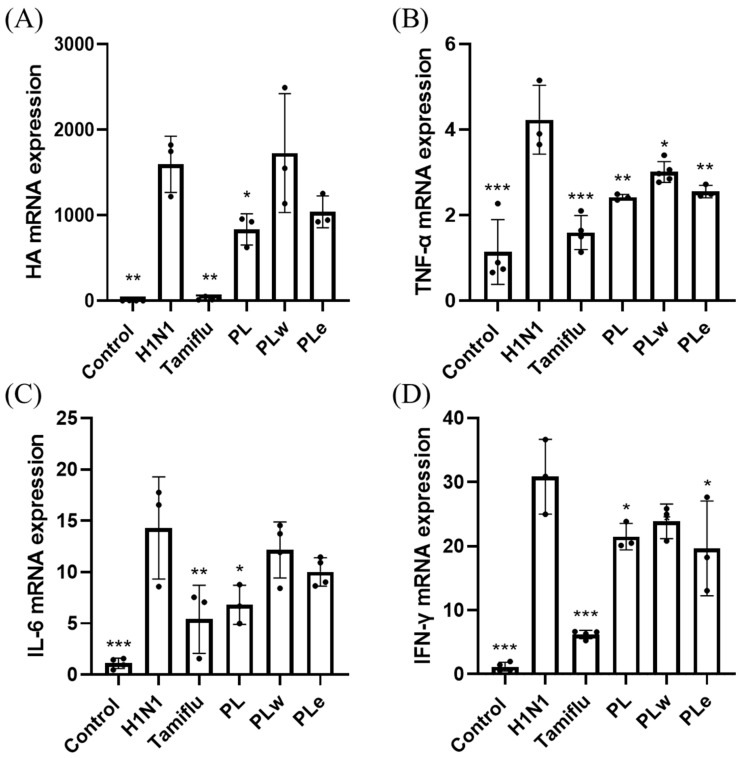
*P. linteus* treatment modulates gene expression in lung tissue of H1N1-infected mice. RT-PCR analysis revealed: (**A**) Viral load assessment via HA expression showed significant reduction with PL treatment; (**B**) TNF-α expression was suppressed across all treatment groups; (**C**) IL-6 expression was significantly decreased by PL treatment; (**D**) IFN-γ levels were notably reduced by both PL and PLe treatments. Values expressed as mean ± SD (n = 4). Statistical significance compared to H1N1-infected group: * *p* < 0.05, ** *p* < 0.01, *** *p* < 0.001.

## Data Availability

The original contributions presented in the study are included in the article/Appendix A, further inquiries can be directed to the corresponding author.

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
