# Peer review of "Phellinus linteus Mycelia Extracts Show Potent Antiviral and Immunomodulatory Effects in H1N1 Influenza Virus-Infected Mice"

_foods, 2025, doi:10.3390/foods14234047_

Round 1
Reviewer 1 Report
Comments and Suggestions for Authors
The presented study aimed to evaluate antiviral and immunomodulatory effects of Phellinus linteus mycelia extract in H1N1 influenza virus infected mice. The topic is important and interesting, although many studies of Phellinus linteus extract are already recorded by other authors. However, this study to some extent may be a useful addition to existing research.
The manuscript is not well prepared and needs to be improved.
Some improvements in MS should be implemented and some questions must be answered:
In the Abstract, please briefly state the methods used in the research. Also, indicate which doses of PL extract were tested.
Please remove Figure 1 from the Materials and Methods section into the Results section. The description of Figure 1 needs to be completed - state the method used in the experiment and the time after which the results were collected. Please check all other figure descriptions and complete them if necessary.
In cell viability assays, please indicate the time after which cells were treated. Also, briefly describe the MTS method.
The Materials and Methods section describes the cultivation of the RAW 264.7 cell line, but in the Results section I do not find any results obtained using this cell culture – please explain.
In the description of Figure 2, there is no mark for p<0.05.
Figure 3 (B) should be displayed more clearly so that the Tamiflu positive control can be seen.
Within the MS text, reference should be made to Figure 5.
Within the MS text, reference should be made to Figure S1 and Figure S3 (Supplementary material).
Line 240, 241 in statement: “PLe exhibits a relatively high potent antiviral activity at concentrations of 250-1000 μg/mL with increased cell viability by 8.23-18.62%” – correct the PLe concentrations
Line 427 – please refer to Figure 3B in the MS text
In flow cytometry analysis for bronchoalveolar lavage fluids (BALF) I don’t see appropriate control (non treated mouse) – please explain
Line 484 – In statement: “improved their survival rates respectively to 60% and 40% when compared to the control group's 25%” - specify the experimental model
Author Response
#Author 1:
The presented study aimed to evaluate antiviral and immunomodulatory effects of Phellinus linteus mycelia extract in H1N1 influenza virus infected mice. The topic is important and interesting, although many studies of Phellinus linteus extract are already recorded by other authors. However, this study to some extent may be a useful addition to existing research. The manuscript is not well prepared and needs to be improved. Some improvements in MS should be implemented and some questions must be answered:
In the Abstract, please briefly state the methods used in the research. Also, indicate which doses of PL extract were tested.
Author’s Reply: We sincerely thank the editor for the suggestion. The methods used in the study have now been briefly described in the Abstract, and the doses of PL powder and extracts have been included (lines 24–29 of the revised manuscript).
Please remove Figure 1 from the Materials and Methods section into the Results section. The description of Figure 1 needs to be completed - state the method used in the experiment and the time after which the results were collected. Please check all other figure descriptions and complete them if necessary.
Author’s Reply: We appreciate the editor’s comment. Figure 1 has been moved from the Materials and Methods section to the Results section. Its description has also been completed by specifying the method used and the time point at which the results were collected (lines 269–276).
In cell viability assays, please indicate the time after which cells were treated. Also, briefly describe the MTS method.
Author’s Reply: We thank the editor for the suggestion. Additional details for the cell viability assay and a brief description of the MTS method have been added to lines 125–133 of the revised manuscript.
The Materials and Methods section describes the cultivation of the RAW 264.7 cell line, but in the Results section I do not find any results obtained using this cell culture – please explain.
Author’s Reply: We apologize for the confusion. This was a typographical error, and the description has been removed from the revised manuscript (lines 114–115).
In the description of Figure 2, there is no mark for p<0.05.
Author’s Reply: We thank the editor for the suggestion. The significance marker (*p < 0.05) has now been added to Figure 2 (line 313).
Figure 3 (B) should be displayed more clearly so that the Tamiflu positive control can be seen.
Author’s Reply: We appreciate the editor’s comment. Figure 3B has been amended to clearly show the Tamiflu positive control.
Within the MS text, reference should be made to Figure 5.
Author’s Reply: References to Figure 5 have been added in lines 368, 369, 372, 377, 378, and 381 as suggested.
Within the MS text, reference should be made to Figure S1 and Figure S3 (Supplementary material).
Author’s Reply: References for Figure S1 have been added in lines 255 and 259, and for Figure S3 in line 317.
Line 240, 241 in statement: “PLe exhibits a relatively high potent antiviral activity at concentrations of 250-1000 μg/mL with increased cell viability by 8.23-18.62%” – correct the PLe concentrations
Author’s Reply: We thank the editor for the suggestion. The PLe concentrations have been corrected in line 285.
Line 427 – please refer to Figure 3B in the MS text
Author’s Reply: Reference to Figure 3B has been added in line 324.
In flow cytometry analysis for bronchoalveolar lavage fluids (BALF) I don’t see appropriate control (non treated mouse) – please explain
Author’s Reply: We thank the editor for the comment. Appropriate non-treated controls have been added to Figure S4.
Line 484 – In statement: “improved their survival rates respectively to 60% and 40% when compared to the control group's 25%” - specify the experimental model1.
Author’s Reply: We thank the editor for the suggestion. The experimental model has been specified in line 544 of the revised manuscript.
Reviewer 2 Report
Comments and Suggestions for Authors
The manuscript presents a comprehensive evaluation of Phellinus linteus mycelial extracts (PL, PLw, PLe) against H1N1 infection in both in vitro and in vivo models. The study is methodologically sound and relevant to the field of functional food research with antiviral potential. However, there are few concerns to address:
- Novelty and dosage rationale
Although the authors state that the aim was to examine the effect of P. linteus mycelial extracts on influenza A virus infection, the novelty is still not fully justified. Similar antiviral studies have already been reported for P. linteus and related species. The manuscript should clearly explain what distinguishes this work, for example, whether it identifies unique bioactive compounds, compares extraction methods systematically, or provides new mechanistic insight.
Additionally, the dosage rationale (1000 mg/kg for PL; 350 mg/kg for extracts) remains unclear. It appears based solely on tolerated doses, not on preliminary efficacy or pharmacokinetic data. Without showing that these doses yield biologically relevant plasma levels or effects, the justification seems incomplete.
- Sex of experimental animals
The in vivo study used only male BALB/c mice. Since immune responses can differ significantly between sexes, the authors should justify excluding females or acknowledge this as a limitation.
- Figure order and reference consistency (Figure 4)
Figure 4 is not properly cited or discussed in section 3.5, even though it directly relates to the immune cell recovery described there.
- Conflict of interest and data transparency
Several authors are affiliated with Grape King Bio Ltd., the producer of P. linteus products. This potential conflict must be explicitly stated. Additionally, raw or summary data (e.g., replicate counts, statistical tables) should be made available to support transparency and reproducibility.
- Line 63 – ACE2 clarification
The text mentions that the influenza virus modulates “ACE2 expression” but does not explain what ACE2 is. For clarity, ACE2 (angiotensin-converting enzyme 2) should be briefly defined in the introduction since it plays a key role in viral entry and lung injury mechanisms.
Author Response
#Author 2: The manuscript presents a comprehensive evaluation of Phellinus linteus mycelial extracts (PL, PLw, PLe) against H1N1 infection in both in vitro and in vivo models. The study is methodologically sound and relevant to the field of functional food research with antiviral potential. However, there are few concerns to address:
Novelty and dosage rationale
Although the authors state that the aim was to examine the effect of P. linteus mycelial extracts on influenza A virus infection, the novelty is still not fully justified. Similar antiviral studies have already been reported for P. linteus and related species. The manuscript should clearly explain what distinguishes this work, for example, whether it identifies unique bioactive compounds, compares extraction methods systematically, or provides new mechanistic insight.
Author’s Reply: We sincerely thank the editor for this important comment. To clarify the novelty of our study, we have added text highlighting how our work differs from previous studies, including the focus on mycelium extracts, identification of bioactive compounds, and exploration of potential synergistic mechanisms (lines 63–72 of the revised manuscript).
Additionally, the dosage rationale (1000 mg/kg for PL; 350 mg/kg for extracts) remains unclear. It appears based solely on tolerated doses, not on preliminary efficacy or pharmacokinetic data. Without showing that these doses yield biologically relevant plasma levels or effects, the justification seems incomplete.
Author’s Reply: We thank the reviewer for this valuable comment. The selected doses were based not only on safety/tolerated levels but also supported by preliminary in vivo experiments in our laboratory. In these experiments, lower doses produced limited or inconsistent antiviral effects, whereas the selected doses consistently showed measurable improvements in clinical symptoms and survival outcomes. We have now clarified this rationale in the revised manuscript (lines 468–472). We also acknowledge that additional pharmacokinetic profiling would further strengthen the justification, which will be addressed in future studies.
Sex of experimental animals
The in vivo study used only male BALB/c mice. Since immune responses can differ significantly between sexes, the authors should justify excluding females or acknowledge this as a limitation.
Author’s Reply: We appreciate this important comment. In our previous H1N1 experiments using BALB/c mice, we observed no significant differences between male and female mice in terms of clinical progression, body weight loss, viral load, or immune responses. Based on this prior evidence, and to reduce unnecessary animal use and variability, we selected male mice for this study. We have also acknowledged this as a limitation in lines 476–479 of the revised manuscript.
Figure order and reference consistency (Figure 4)
Figure 4 is not properly cited or discussed in section 3.5, even though it directly relates to the immune cell recovery described there.
Author’s Reply: We apologize for the oversight. References to Figure 5 have now been added in lines 368, 369, 372, 377, 378, and 381.
Conflict of interest and data transparency
Several authors are affiliated with Grape King Bio Ltd., the producer of P. linteus products. This potential conflict must be explicitly stated. Additionally, raw or summary data (e.g., replicate counts, statistical tables) should be made available to support transparency and reproducibility.
Author’s Reply: Thank you for raising this point. Conflict of interest and data transparency statements have been included in the revised manuscript.
Line 63 – ACE2 clarification
The text mentions that the influenza virus modulates “ACE2 expression” but does not explain what ACE2 is. For clarity, ACE2 (angiotensin-converting enzyme 2) should be briefly defined in the introduction since it plays a key role in viral entry and lung injury mechanisms.
Author’s Reply: We thank the editor for the suggestion. ACE2 has now been briefly defined as “angiotensin-converting enzyme 2, a receptor involved in viral entry and lung injury regulation” in lines 77–78 of the revised manuscript.
Round 2
Reviewer 1 Report
Comments and Suggestions for Authors
Please specify the instrument on which the absorbance is determined in the MTS test
Author Response
Please specify the instrument on which the absorbance is determined in the MTS test
Author’s Reply: We thank the reviewer for this helpful comment. The instrument used to determine the absorbance in the MTS test—an Agilent BioTek Epoch 2 Microplate Spectrophotometer—has been added to lines 131–132 of the revised manuscript.